# G-Protein-Coupled Estrogen Receptor (GPER)-Specific Agonist G1 Induces ER Stress Leading to Cell Death in MCF-7 Cells

**DOI:** 10.3390/biom9090503

**Published:** 2019-09-18

**Authors:** Diep-Khanh Ho Vo, Roland Hartig, Sönke Weinert, Johannes Haybaeck, Norbert Nass

**Affiliations:** 1Department of Pathology, Otto-von-Guericke University Magdeburg, Leipziger Str. 44, D-39120 Magdeburg, Germany; johannes.haybaeck@med.ovgu.de; 2Institute of Molecular and Clinical Immunology, Otto-von-Guericke University Magdeburg, Leipziger Str. 44, D-39120 Magdeburg, Germany; roland.hartig@med.ovgu.de; 3Department of Cardiology, Medical Faculty, Otto-von-Guericke University Magdeburg, Leipziger Str. 44, D-39120 Magdeburg, Germany; soenke.weinert@med.ovgu.de; 4Department of Pathology, Neuropathology, and Molecular Pathology, Medical University of Innsbruck, Innrain, Christoph-Probst-Platz 52, D-6020 Innsbruck, Austria; 5Department of Pathology, Diagnostic & Research Center for Molecular BioMedicine, Institute of Pathology, Medical University of Graz, Auenbruggerpl. 2, D-8036 Graz, Austria

**Keywords:** GPER, GPER-specific agonist G1, UPR signaling, Ca^2+^ efflux, ER stress, JNK, CAMKII, breast cancer

## Abstract

The G-protein-coupled estrogen receptor (GPER) mediates rapid non-genomic effects of estrogen. Although GPER is able to induce proliferation, it is down-regulated in breast, ovarian and colorectal cancer. During cancer progression, high expression levels of GPER are favorable for patients’ survival. The GPER-specific agonist G1 leads to an inhibition of cell proliferation and an elevated level of intracellular calcium (Ca^2+^). The purpose of this study is to elucidate the mechanism of G1-induced cell death by focusing on the connection between G1-induced Ca^2+^ depletion and endoplasmic reticulum (ER) stress in the estrogen receptor positive breast cancer cell line MCF-7. We found that G1-induced ER Ca^2+^ efflux led to the activation of the unfolded protein response (UPR), indicated by the phosphorylation of IRE1α and PERK and the cleavage of ATF6. The pro-survival UPR signaling was activated via up-regulation of the ER chaperon protein GRP78 and translational attenuation indicated by eIF2-α phosphorylation. However, the accompanying pro-death UPR signaling is profoundly activated and responsible for ER stress-induced cell death. Mechanistically, PERK-phosphorylation-induced JNK-phosphorylation and IRE1α-phosphorylation, which further triggered CAMKII-phosphorylation, are both implicated in G1-induced cell death. Our study indicates that loss of ER Ca^2+^ is responsible for G1-induced cell death via the pro-death UPR signaling.

## 1. Introduction

Estrogens have a multitude of cellular and physiological effects, ranging from the control of reproduction to the regulation of cell differentiation and proliferation. Besides canonical genomic signaling via nuclear estrogen receptors [1], there are signaling pathways comprising membrane bound estrogen receptors such as the G-protein-coupled estrogen receptor (GPER), formerly named GPR30 [2,3,4]. GPER mediates the rapid non-genomic effects of estrogen, including the production of cyclic adenosine monophosphate (cAMP), the mobilization of intracellular calcium (Ca^2+^), transactivation of epidermal growth factor receptor (EGFR) and the activation of multiple kinases, such as phosphoinositide 3-kinase/protein kinase B (PI3K/AKT) and ERK1/2 mitogen activated-protein kinases (MAPK) [5], often resulting in increased proliferation. GPER is located not only on the plasma membrane but also on the membrane of the endoplasmic reticulum (ER) [4], and is found to be ubiquitously expressed in the body [5,6]. 

From a clinical point of view, GPER is downregulated with breast cancer (BC) tumor progression and a high expression correlates with favorable patient survival [7,8,9,10]. A similar observation was also reported for ovarian [11] and colorectal cancer (CRC) [12]. Nevertheless, high GPER expression also correlated with the development of distant metastasis in BC [13]. In case of BC anti-estrogen therapy, high GPER levels were associated with a shorter disease-free survival under tamoxifen when compared to those receiving aromatase inhibitors [14]. These conflicting observations are in line with the idea that non-specific GPER agonists, such as estrogen and tamoxifen, induce cell proliferation by GPER stimulation, thereby replacing the tamoxifen-blocked genomic estrogen signaling and activating a cross talk of GPER with growth factor signaling [15,16].

Nonsteroidal, high-affinity, highly selective agonists of GPER, such as G1, have been developed from a library of 10,000 molecules [17]. G1 selectively activates GPER but not the classical estrogen receptors. Moreover, previous reports showed that G1 profoundly inhibited cell proliferation and induced apoptosis in BC, prostate cancer and CRC cell lines [12,18,19,20,21,22], potentially via p53 and p21 induction [19,23]. G1-mediated GPER activation also elicited cytosolic Ca^2+^ currents in BC cell lines [18]. Decreasing GPER abundance by siRNA inhibited both the upregulation of the L-type Ca^2+^ channel subunit α 1D named Cav1.3, and the 17β-estradiol (E_2_)-induced Ca^2+^ influx. The inhibition of Cav1.3 by siRNA suppressed E_2_-induced intracellular Ca^2+^ upregulation and cell proliferation [24]. However, little is known about the precise apoptotic mechanism following the mobilization of intracellular Ca^2+^ induced by GPER-specific agonist G1 in BC. In vivo, G1 inhibited tumor growth in estrogen receptor (-) BC cells [23] and HCT116 CRC tumor xenografts in mice [12]. These studies suggest that GPER can serve as a potential therapeutic target in BC. 

Ca^2+^ homeostasis is important for the functioning of ER as ER-resident chaperones, such as calreticulin and glucose-regulated protein 78 (GRP78), which need a high concentration of Ca^2+^ for their activity [25]. If ER Ca^2+^ homeostasis is lost, the unfolded protein response (UPR) will be activated, either re-establishing the ER functions or triggering cell death. The UPR is mediated by three major signaling proteins, inositol-requiring enzyme 1α (IRE1α), protein kinase R-like ER kinase (PERK, EIF2AK3) and activating transcription factor 6 (ATF6) [26]. If ER Ca^2+^ is severely decreased, ER stress-induced cell death will be promoted by either affecting mitochondrial Ca^2+^ via causing mitochondrial outer membrane permeabilization or by the loss of the mitochondrial transmembrane potential (ΔΨm) [27], or by activating the pro-death UPR signaling. These events can directly release cytochrome c to the cytosol to induce apoptosis.

Mechanistically, ER stress stimulates the autophosphorylation of the IRE1α cytosolic domain to become an active endoribonuclease (RNAse) cleaving an intron of *X-box binding protein 1* (*XBP1*) mRNA, resulting in the translation of bZIP-containing transcription factor XBP1s [28]. XBP1s activates the adaptive response by inducing the transcription of genes to assist ER stress. Activated IRE1α also causes phosphorylation of Jun-N-terminal kinase (JNK) and p38-MAPK to promote apoptosis [29]. On activation, autophosphorylated PERK can either induce pro-survival or pro-apoptotic signaling. Activated PERK catalyzes the phosphorylation of eukaryotic translation initiator factor 2α (eIF2α) to attenuate protein biosynthesis. In addition, phosphorylation of eIF2α also induces the translation of activating transcription factor 4 (ATF4) [26]. On activation by the binding of GRP78, ATF6 translocates to the Golgi apparatus, where it is cleaved to become an active transcription factor and to further activate transcription of target genes such as ER chaperones [26]. Both ATF6 and ATF4 also control the expression of pro-apoptotic proteins, including C/EBP-homologous protein (CHOP) [26]. p38 is also reported to activate CHOP, in turn causing changes in gene expression that favor apoptosis [30]. Therefore, CHOP is regarded as one of the most important mediators of ER stress-induced apoptosis [31].

The approach of Ca^2+^ depletion-induced ER stress leading to cell death has been used to develop new chemotherapies, for example thapsigargin and curcumin, two well-known sarco/ER Ca²⁺-ATPase (SERCA) blockers [32,33]. In the present study, we investigated the precise connection between GPER activation by G1 and ER stress causing cell death in the estrogen receptor-positive BC cell line MCF-7. We found that G1-induced Ca^2+^ efflux activates the UPR; however, up-regulation of CHOP or the mitochondrial transmembrane potential is not involved in G1-induced cell death. We further demonstrate that the phosphorylation of PERK, which directly induces the activation of JNK kinase, is implicated in G1-induced cell death. Moreover, the phosphorylation of IRE1α does not cleave *XBP1* but directly activates calcium/calmodulin-dependent protein kinase II (CaMKII), causing G1-induced cell death. We conclude that G1 triggers a mobilization of ER Ca^2+^ stores, leading to UPR activation. The accompanying pro-death UPR signaling is then responsible for G1-induced cell death

## 2. Materials and Methods

### 2.1. Reagents

G1 was purchased from Tocris (Wiesbaden-Nordenstadt, Germany), dissolved (5 mM) in dimethyl sulfoxide (DMSO) (Roth, Karsruhe, Germany) and stored at −20 °C; Thapsigargin, SP600125, GSK2606414 were also purchased from Tocris. Indo-1 AM was from Thermo Fisher Scientific (Waltham, MA, USA). zVAD-fmk was bought from Santa Cruz (Santa Cruz, CA, USA). SB203580 and Kira6 were purchased from MERCK Millipore (Darmstadt, Germany). All substances were dissolved in DMSO. Antibodies were obtained from the following commercial sources: caspase 9 (Ca# 9502), cleaved PARP (Ca# 9541), IRE1α (Cat# 3294), PERK (Cat# 3192), eIF2α (Cat# 5234), phospho-eIF2α (Cat# 3398), BiP GRP78 (Ca# 3177), CHOP (Cat# 2895), p38 MAPK (Ca# 9212), phospho-p38 MAPK (Ca# 4511), phospho-SAPK/JNK (Ca# 4668), caspase 3 (Ca# 9662), BCL-2 (Ca# 2872), Cell Signaling (Danvers, MA, USA); ATF6 (Cat# 73500), BioAcademia (Osaka, Japan); puromycin (Ca# MABE343), cylophilin D (Ca# AP1035), MERCK Millipore (Darmstadt, Germany); phospho-IRE1α (Cat# NBP2-50067), Novus Biologicals (Littleton, CO, USA); cytochrome c (Ca# 556433), BD Biosciences (Franklin Lakes, NJ, USA); β-actin (Cat# A5441, Sigma-Aldrich (Steinheim, Germany)). Secondary, peroxidase-conjugated antibodies were purchased from Dianova (Hamburg, Germany). All other chemicals of analytical grade were obtained from Sigma-Aldrich or Roth.

### 2.2. Cell Lines and Cell Culture

MCF-7 cells were obtained from the American Type Culture Collection (ATCC, HTB-22) (Manassas, VA, USA). Cells were routinely maintained in phenol-red-free RPMI 1640, which contained 10% fetal bovine serum (FBS) and 200 µM L-glutamax (all from Biochrom, Berlin, Germany). Cells were grown at 37 °C in an atmosphere of 95% air and 5% CO_2_ and transferred into new flasks (Nunc) after detachment with Trypsin/EDTA (Biochrom).

### 2.3. Cell Treatment

To elucidate the mechanism of cell death induced by GPER-specific agonist G1 via ER stress, MCF-7 cells were treated with 1, 2.5 and 5 µM G1 for the indicated period in growth medium containing FBS. As positive controls, cells were also exposed to 1 µM thapsigargin for the indicated time. DMSO was used as a vehicle for control treatments. To evaluate the effect of pan caspases inhibitor zVAD-fmk, cells were pretreated with 20 µM zVAD for 1 h before further treatment. Cells were also pretreated with a variable concentration of kinase inhibitors SB203580, SP60025, GSK2606414 and Kira6 for 1 h before further treatment.

### 2.4. Cell Cycle and Apoptosis Analysis by Flow Cytometry

MCF-7 cells were collected 24, 48 and 72 h after treatment. For cell cycle analysis, cells were fixed with 70% ethanol, treated with 1% RNase in TE buffer and finally stained with a hypotonic propidium iodide (PI) solution (50 µg/mL in PBS). Cell cycle analysis was performed using a flow cytometer (LSRFortessa, BD Bioscience, San Jose, CA, USA). Cell cycle distribution (percentage of cells) in cell debris (sub-G1) and G1, S, and G2/M phases of the cell cycle was analyzed using FlowJo software version 7.6 (Treestar, Ashland, OR, USA). To discriminate between apoptosis and necrosis, cells were detached with Trypsin/EDTA and stained with FITC Annexin V (BioLegend, San Diego, CA, USA) and PI (50 µg/mL). Then, necrotic and apoptotic cells were determined using a flow cytometer (LSRFortessa). Data were analyzed using FlowJo software version 7.6 (Treestar, Ashland, OR).

### 2.5. Immunoblotting 

A whole cell extract was prepared as described previously [34]. Briefly, cells were suspended in a lysis buffer (150 mM NaCl, 50 mM Tris-HCl (pH 7.5), 1% NP-40, 1% sodium deoxycholate, 0.1% SDS) containing phosphatase and protease inhibitor cocktails (MERCK Millipore, Darmstadt, Germany) on ice for 30 min. Cellular debris was removed by centrifugation for 15 min at 12,000× *g* and 4 °C. Protein content was determined by the BCA reagent (Thermo Fisher Scientific (Waltham, MA, USA)), and equal amounts were subjected to denaturing SDS-PAGE. Blotting was performed with a BioRad semi-dry blotting chamber using a nitro cellulose membrane (GE Healthcare, Chicago, Illinois, USA) for 1 h at 1 mA per cm^2^ in CAPS buffer (CAPS 50 mM pH 10, methanol 10% and β-mercapto-propionic acid 1 mM). Transfer to the membrane was controlled by Ponceau red staining, and the membrane was blocked with 2% BSA or 5% skim milk in TBS (Tris/Cl 50 mM pH 7.4, NaCl 150 mM, 0.2% NP40). Immunoblotting with appropriate antibodies was visualized with an enhanced chemiluminescence reagent (MERCK Millipore, Darmstadt, Germany) in an INTAS chemstar imager (Intas, Goettingen, Germany).

### 2.6. Indo-1 AM Staining 

To measure cytosolic Ca^2+^, MCF-7 cells were stained with 1 μM Indo-1-AM (Thermo Fisher Scientific) in RPMI1640 medium (phenol-red free) containing 10% FBS at 37 °C for 45 min. Then, cells were washed and incubated for another 45 min at 37 °C in RPMI 1640 containing 10% FBS. After washing, the samples were analyzed using a flow cytometer (LSR I BD Bioscience, Heidelberg, Germany). Cells were illuminated with the 325 nm laser line of a helium–cadmium laser. Flourescence emissions at 390 to 420 nm and 500 to 520 nm were detected simultaneously, and changes in the ratio of the two emission intensities were analyzed with FLowJo software. To demonstrate successful loading with the dye, we induced maximal Ca^2+^ release by adding calcium ionophore ionomycin (10 mg/mL) (Sigma-Aldrich). Ca^2+^ efflux in MCF-7 cells was performed in response to varying concentrations of G1 or TG. DMSO was used as a negative control. 

### 2.7. Measurement of Protein Synthesis by the Surface Sensing of Translation (SUnSET) Puromycin End-Labeling Assay 

At the end of a treatment, cells were incubated with 10 µg/mL puromycin for 15 min prior to cell lysis. Whole cell lysates were then subjected to immunoblotting. Anti-puromycin antibody was used to detect the level of puromycin-labeled proteins by Western blotting as described above.

### 2.8. Real-Time PCR

To determine *CHOP* mRNA levels, quantitative RT-PCR was carried out. RNA isolation, cDNA synthesis and quantitative (Q) PCR were performed as follows: The RNA isolation kit was purchased from Macherey and Nagel, (Düren, Germany). Briefly, 1 μg total RNA was used for each cDNA synthesis by using BioScript Reverse Transcriptase (Bioline, Germany) using oligo dT primers (Promega, Madison, Wisconsin, USA) and random hexamer primers (Promega) separately. Both reactions were then pooled and used for qPCR. The PCR reaction was monitored in a Roche light cycler 2.0 by using the SYBR Green I Master (Roche, Mannheim, Germany) added to the PCR reaction mix. The relative mRNA level of *CHOP* was calculated by the 2^−ΔΔ*Ct*^ method using human *RPL13A* as the reference gene for normalization. The following primers (Biomers, Germany) were used: human *CHOP* forward, 5′-agtctaaggcactgagcgta-3′; reverse, 5′-ttgaacactctctcctcaggt-3′; human *RPL13A* forward, 5′-cctggaggagaagaggaaagaga-3′; reverse, 5′-ttgaggacctcttgtgtatttgtcaa-3′. 

### 2.9. Knockdown of CHOP by Small Interfering RNA

ON-TARGETplus DDIT3 (CHOP) siRNA was obtained from GE-Dharmacon (Colorado, USA). ON-TARGET*plus* Non-targeting Control Pool (GE-Dharmacon) was used as negative control for RNAi experiments. The siRNA was transfected into MCF-7 cells at a final concentration of 25 nM siRNA by Dharmafect 2 transfection reagent (GE-Dharmacon) for 24 h before further experiments. To confirm the knockdown efficiency for CHOP, mRNA expression and protein content were analyzed by qPCR and immunoblotting.

### 2.10. Determination of Cell Viability

For determination of cell viability, the lactate dehydrogenase (LDH) assay or resazurin assay was used. The LDH activity assay was performed as described previously [34]. Data were expressed as the percentage of total LDH activity, after subtraction of the background determined from the unused culture medium alone. Alternatively, resazurin (10 µg/mL) was added to the cell culture medium and incubated for 30 to 120 min at 37 °C. Then 100 µL resazurin solution was sampled and fluorescence recorded at wavelengths of 525/580–640 nm in a CLARIOstar microplate reader (BMG LabTech, Ortenberg, Germany).

### 2.11. Mitochondrial Potential Measurement 

To measure the mitochondrial membrane potential (ΔΨm), MCF-7 cells were stained with 10 nM tetramethylrhodamine ethyl ester (TMRE) (BD Pharmingen, San Diego, CA, USA)) in RPMI1640 medium (phenol-red free) containing 10% FCS at 37 °C for 15 min. MitoStatus TMRE is a fluorescent dye that is readily sequestered by active mitochondria, allowing flow cytometric or imaging analysis to assess apoptosis or mitochondrial depolarization. Then, cells were washed 2 times with cold RPMI medium. After washing, the samples were analyzed using a flow cytometer (LSRFortessa). Values of Δψm were analyzed using FlowJo software. Δψm was determined in response to 1 µM G1 or TG at 3, 5, 8, 16 and 24 h. DMSO was used as a negative control. Carbonyl cyanide *m*-chlorophenyl hydrazone (CCCP) was used as a positive control. It induces the opening of the permeability transition pore on the mitochondrial membrane, leading to the dissipation of ΔΨm.

### 2.12. Subcellular Fractionation

Cells were collected by gentle scratching in a 200 μL digitonin buffer (150 mM NaCl, 10 mM Tris-HCl (pH 7.4), 40 μg/mL digitonin (Roth, Karlsruhe, Germany)). After 10 min incubation on ice, the cell suspension was centrifuged at 8000× *g* for 5 min at 4 °C. The resulting supernatant was kept as a cytosolic-enriched digitonin extract. The resulting pellet was resuspended by vortexing in 200 μL NP-40 buffer (150 mM NaCl, 10 mM Tris-HCl (pH 7.4), 1% NP-40). After 30 min of incubation on ice, the cell pellet was centrifuged at 13,000× *g* for 5 min at 4 °C. The resulting supernatant was kept as a membrane organellar protein-enriched NP-40 extract. Equal aliquots from each fraction were analyzed for immunoblotting.

### 2.13. Statistical Analysis

Data are reported as mean ± SD of at least three independent experiments unless otherwise indicated. The statistical significance of the difference between the determinations was calculated by analysis of variance using ANOVA, a Tukey–Kramer multiple comparisons test or a Student’s *t* test. The difference was considered as significant when the *p* value was <0.05 or <0.01.

## 3. Results

### 3.1. G1 and TG Induced Cell Cycle Arrest in G2/M and Apoptosis without Activating Caspase-3 and 9 in MCF-7 Cells

G1 was proposed to induce apoptosis in MCF-7 cells, but the detailed mechanism remains elusive [18,22]. We analyzed this process further to better understand the molecular pathway leading to G1-induced cell death.

MCF-7 cells were treated with 1 μM G1 or 1 μM G1 thapsigargin (TG), a procedure that was reported to induce apoptosis in MCF-7 cells [35]. Cell cycle and apoptosis were re-evaluated after 24, 48 and 72 h by flow cytometry. Our results showed that G1 inhibited proliferation of MCF-7 cells by a remarkable arrest in G2/M already 24 h after stimulation compared to the control group (Figure 1a). This block was associated with a concomitant decrease in the percentage of cells in the G1 phase, and a smaller reduction in the S phase. This effect was also evident after 48 and 72 h of G1 treatment. TG stimulation showed a decrease in cells in the S phase after 24 h and in the G2/M checkpoint after 48 h (Figure 1a). These results indicate that the stimulation of GPER by G1 prevents cells from entering both the synthesis phase and the mitotic phase, thereby causing an inhibition of cell proliferation of BC MCF-7 cells. 

Next, we re-investigated the characteristics of apoptosis in MCF-7 cells induced by G1. After treatment, cells were stained with Annexin V and propidium iodide (PI), and apoptotic cells were determined by flow cytometry. Both G1 and TG treatment led to early apoptosis in a small number of cells, while about 10% of the cells showed signs of late apoptosis after 24 h. This amount of late apoptotic cells was further increased at 48 h and remained constant at about 18% at 72 h compared to the control group (Figure 1b,c), indicating a constant rate of apoptosis after 48 h resulting in increasing numbers of dead detached cells. Notably, expression of caspase 3 is completely absent in MCF-7 cells compared to SK-BR-3 and MDA-MB-231 cell lines (Appendix A). We observed the cleavage of PARP but not of caspase 9 after 24 h stimulation by G1 and TG in all concentrations tested: 1, 2.5 and 5 μM compared to the DMSO treatment (Figure 1d). Western Blot (WB) revealed similar results after 48 and 72 h treatment. Then, we examined the effect of the pan-caspase inhibitor zVAD-fmk (zVAD) on G1- and TG-induced apoptosis in MCF-7 cells. WB analysis showed that the cleavage of PARP was inhibited in the presence of zVAD compared to the control cells (Figure 1d). Due to the intracellular release of toxic fluoroacetate [36], zVAD-fmk could not significantly prevent G1-induced apoptosis (data not shown). Taken together, these results confirm that after 24 h stimulation, GPER-specific agonist G1 induces cell cycle arrest at G2/M and late apoptosis although caspases 3 and 9 are not involved in the mechanism.

### 3.2. G1 and TG Induced Ca^2+^ Efflux and Activated the UPR in MCF-7 Cells

As previous studies have shown that G1 resulted in Ca^2+^ mobilization from the ER [18], we hypothesized that loss of Ca^2+^ of the ER could contribute to the cell death mechanism. As a first step in investigating this process further, we re-investigated this effect by using the Indo-1 dye as indicator to analyze the intracellular Ca^2+^ concentration. Figure 2a revealed an elevation of Ca^2+^ concentration in the cytosol with G1 concentrations at 0.5, 1 and 5 μM compared to the DMSO treatment. A similar but much stronger and faster increase of Ca^2+^ efflux from the ER was detected in TG-stimulated MCF-7 cells (Figure 2b). TG is a well-established ER stress inducer known to inhibit ER Ca^2+^-ATPase. We also blocked extracellular Ca^2+^ influx by using EGTA, a chelating agent that has a high affinity to Ca^2+^. EGTA showed no effects on the increase in Ca^2+^ influx (data not shown), indicating that the mobilization of Ca^2+^ induced by G1 comes from intracellular sources, presumably the ER. The results of previous and our own research led us to conclude that stimulation by G1 leads to a rapid Ca^2+^ mobilization from the ER to the cytosol, and this could play an important role in G1-induced cell death. 

The loss of Ca^2+^ from the ER led us to hypothesize that ER stress might be induced by and involved in the mechanism of G1-induced MCF-7 cell death. Therefore, we examined the activation of the UPR by immunoblotting. The phosphorylation of IRE1α was induced by all G1 concentrations (1, 2.5 and 5 μM) tested after 24, 48 and 72 h treatment (Figure 2c). The same result was observed in cells treated with 1 μM TG for 24 and 48 h. However, G1-induced IRE1α phosphorylation could not induce XBP1 splicing (XBP1s) compared to TG-activated IRE1α phosphorylation (Figure 2c). Moreover, G1 and TG promoted the reduction of full-length ATF6 in an equivalent experimental setting as compared with DMSO-treated MCF-7 cells (Figure 2c). The phosphorylation of PERK was also induced in cells treated with all G1 concentrations tested. This became evident in the form of a mobility shift of PERK bands on SDS-PAGE as compared with the control cells (Figure 2c). The same band shifts were observed in cells treated with 1 μM TG. We also found that the expression of ER-resident proteins, such as GRP78, was elevated in G1 and TG-treated cells as compared with the control group (Figure 2c). Altogether, our findings indicate that the UPR is indeed activated, and ER stress occurs upon GPER stimulation by G1.

We further found that the phosphorylation of eIF2α was induced in both G1 and TG treatment as compared with DMSO treatment in MCF-7 cells. Using SUnSET, a non-radioactive system monitoring newly synthesized proteins based on probing puromycin-labeled proteins [28,37], we observed a decrease in puromycin-labeled proteins in the cells treated with 1 μM of G1 for 24, 48 and 72 h as compared with the control group (Figure 2d). These results suggest that stimulation of GPER by its specific agonist G1 also inhibits global protein synthesis via the phosphorylation of eIF2α in MCF-7 cells. 

We then hypothesized that increased Ca^2+^ concentration in the cytosol due to GPER stimulation by G1 could lead to a functional loss of the mitochondrial potential, resulting in apoptosis. To elucidate this hypothesis, we stained MCF-7 cells with tetramethylrhodamine ethyl ester (TMRE), which is a positively charged dye that binds to internal negative charges of the mitochondrial inner membrane. The intensity of TMRE staining of the cells represents the mitochondrial transmembrane potential (ΔΨm). Carbonyl cyanide m-chlorophenyl hydrazone (CCCP) was used as a positive control as it induces the opening of the permeability transition pore on the mitochondrial membrane, leading to the dissipation of ΔΨm [38]. However, in our experiments, ΔΨm was not affected by G1 nor the solvent-control (DMSO) (Appendix A), whereas CCCP decreased the ΔΨm level, indicating mitochondrial functional loss. During apoptosis, cytochrome c is released from the intermembrane compartment of mitochondria to the cytosol to activate pro-caspase 9. Because pro-caspase 9 was not activated in G1- and TG-treated cells (Figure 1d), we hypothesized that cytochrome c was not released by G1 and TG either. To clarify this, we performed crude subcellular fractionation to sequentially enrich cytosolic proteins in a digitonin-soluble fraction and membrane-bound organelles in a subsequent NP-40-soluble fraction. Staurosporine (STS) was used as a positive control to release cytochrome c from mitochondria to cytosol (Appendix A). The results showed that the abundance of cytochrome c was not released from the NP-40 organellar fraction into the digitonin cytosolic fraction in both G1- and TG-treated cells (Appendix A). Nevertheless, such a release was evident in STS treated cells. However, the cytosolic house-keeping protein β-actin was also found in the digitonin extract under all treatments, indicating a contamination of this fraction by cytosolic proteins. These results suggest that the mitochondrial transmembrane potential is not affected and cytochrome c is not released upon G1-induced ER Ca^2+^ mobilization.

### 3.3. Up-Regulation of CHOP Expression Was Not Involved in G1-Induced Cell Death in MCF-7 Cells

To investigate whether G1-induced UPR activated downstream pro-death signaling, we studied the involvement of CHOP. The mRNA and protein levels of CHOP were increased by G1 typically after 48 and 72 h treatment compared to the control group (Figure 3a,b). Similar results were observed in TG-treated cells (Figure 3b). siRNA-mediated reduction of CHOP could inhibit the expression of CHOP induced by G1 and TG but not the cleavage of PARP (Figure 3c). CHOP siRNA also failed to prevent cell death induced by G1 and TG in MCF-7 cells (Figure 3d,e). These data suggest that CHOP induction is not involved in the mechanism of G1-induced cell death. 

### 3.4. G1 Activated p38 and JNK and a Selective Inhibitor of Phosphorylation of JNK Inhibited G1-Induced Cell Death in MCF-7 Cells

We then examined whether the activation of p38-MAPK and JNK was implicated in G1-induced ER stress and cell death. Both p38 and p46-JNK, as well as p54-JNK were phosphorylated as a response to all tested concentrations of G1 in a 24-h treatment, and the signal became more prominent after 72 h of treatment compared to the control group. Similar results were obtained with TG-treated cells (Figure 4a), whereas the expression levels of total proteins p38 and JNK were not affected by G1 and TG. We then evaluated the effects of the selective kinase inhibitors SB203580 and SP600125, which inhibit the phosphorylation of p38 and JNK, respectively. While SB203580 suppressed the phosphorylation of p38 induced by G1, the inhibitor could not prevent G1-induced cell death as shown by cell viability analysis and WB results of cleaved-PARP (Figure 4b,c). On the other hand, SP600125 suppressed the phosphorylation of p46-JNK and p54-JNK and significantly inhibited G1-induced cell death (Figure 4d,e). Notably, protein cleavage of PARP was not reduced in cells co-incubated with G1 and SP600125 and rather enhanced in cells treated with SP600125 alone (Figure 4d). This could be the consequence of the self-toxicity of SP600125 on MCF-7 cells, which was shown in cell viability results. Thus, these results indicate that JNK activation plays an important role in the mechanism of G1-induced apoptosis in BC MCF-7 cells.

### 3.5. Inhibition of PERK Inhibited the Activation of JNK and Prevented G1-Induced Cell Death in MCF-7 Cells

We then evaluated whether the activation of PERK could lead to the activation of p38 and JNK in G1-treated cell death. MCF-7 cells were pretreated with 5 μM GSK2606414, an inhibitor of PERK kinase, for 1 h and then exposed to 1 μM G1 or TG for 24, 48 and 72 h. Figure 5a shows that in all cases, the phosphorylation of PERK was suppressed in the presence of GSK2606414. GSK2606414 also inhibited the phosphorylation of both p46-JNK and p54-JNK, as well as p38, and further reduced the cleavage of PARP in G1- and TG-treated cells. Moreover, co-treatment with 5 μM of GSK2606414 significantly prevented G1-induced cell death and slightly mitigated TG-induced cell death (Figure 5b,c), suggesting that p-PERK-mediated p-JNK is directly involved in the mechanism of G1-induced cell death. We also examined the effect of GSK2606414 on the phosphorylation of eIF2α and a newly synthesized protein. GSK2606414 itself seemed to activate p-eIF2α and therefore reduced the cap-dependent nascent protein synthesis in vehicle treatment. Following the co-treatment of G1 and GSK2606414, the phosphorylation of eIF2α was reduced compared to G1 treatment alone (Figure 5a). These results point out that the phosphorylation of PERK by G1 not only regulates translation attenuation but also directly activates JNK to induce apoptosis in BC MCF-7 cells.

### 3.6. Inhibition of IRE1α Inhibited the Phosphorylation of CAMKII and Prevented G1-Induced Cell Death in MCF-7 Cells

The results shown above raised the question of whether activation of IRE1α is involved in G1-induced MCF-7 cell death by joining PERK to activate the phosphorylation of p38 and JNK. We used Kira6, an inhibitor of the IRE1α autophosphorylation, to clarify this hypothesis. A total of 0.05 μM of Kira6 showed inhibitory effects on the autophosphorylation of IRE1α after 48 and 72 h exposure of G1 (Figure 6a). Unlike GSK2606414, Kira6 could not suppress the phosphorylation of JNK but that of p38. Notably, the cleavage of PARP induced by G1 was not changed by Kira6. These results suggest that the phosphorylation of IRE1α does not activate phosphorylation of JNK. Therefore, we investigated whether CAMKII was activated due to the increase in G1-induced Ca^2+^ efflux and played a role as a downstream signal of IRE1α-phosphorylation. Both the protein expression levels and phosphorylation of CAMKII were increased in all concentrations of G1 tested after 24, 48 and 72 h (Figure 6b). Notably, antibody against Thr-286-phosphorylated CAMKII showed two bands at 50 kDa and 60 kDa, indicating phosphorylation of CAMKIIα and CAMKIIβ, respectively. Furthermore, Kira6 could inhibit the phosphorylation of CAMKII (Figure 6a) and significantly inhibited G1-induced cell death (Figure 6c). These results indicate that p-IRE1α-mediated phosphorylation of CAMKII plays a role in G1-induced cell death in MCF-7 cells. 

## 4. Discussion

GPER is highly expressed in a variety of tissues and often involved in estrogen-dependent diseases. GPER expression is down-regulated in various cancers, and high expression is associated with improved survival of patients with breast and ovarian cancers [7,8,10,11]. On the other hand, GPER activation using classical estrogen receptor-interacting ligands, such as E_2_, tamoxifen and 4-OH tamoxifen, was reported to promote cell proliferation of both epithelial and carcinogenic breast cell lines [39,40]. Notably, unlike E_2_, G1, a quinolone derivative, shows nearly no affinity for estrogen receptor-α and -β [17], thus specifically activating GPER. Many research groups have demonstrated that GPER stimulation with high doses of G1, often in the micromolar range, leads to inhibition of growth and proliferation of several BC cell lines [18,21,22]. Although such concentrations are well above the reported IC50 value (3–6 nM), siRNA mediated reduction of GPER expression showed that this inhibitory effect indeed depends on GPER [23]. In models of estrogen receptor (-) BC cells [23] and HCT116 CRC cells in tumor xenograft-bearing mice [12], G1 inhibits tumor growth. In GPER-positive patients, tamoxifen treatment was associated with a significantly shorter disease-free survival compared to those receiving aromatase inhibitors [14]. This indicates that GPER might be involved in tamoxifen resistance of BC. 

A previous study has shown that knockdown of GPER in MCF-7 cells but not of estrogen receptor α blocked E_2_ and G1-induced Ca^2+^ mobilization [18]. This result indicated that the mobilization of intracellular Ca^2+^ stores is dependent on GPER. Moreover, the increase in the intracellular concentration of Ca^2+^ upon stimulation of GPER by G1 has also been reported in estrogen receptor (-) BC cells and cancer-associated fibroblasts [40]. Little is known about a link between the mobilization of intracellular Ca^2+^ stores and cell death when GPER is stimulated by G1. As the ER lumen is the major source of intracellular Ca^2+^, loss of Ca^2+^ can promote a variety of signaling mechanisms that might lead to cell death. The present study shows that G1-induced Ca^2+^ efflux activates the UPR in MCF-7 cells accompanied by the activation of IRE1α, PERK and ATF6. It is likely that a pro-survival UPR is activated by upregulation of GRP78 to assist an accumulation of unfolded proteins in the ER lumen and phosphorylation of eIF2α to reduce translation of nascent proteins. We further show that G1-induced ER Ca^2+^ mobilization prominently activates the pro-death UPR signaling pathways. CHOP is upregulated but not involved in the G1-induced cell death machinery. Unlike the signaling observed in previous studies [29], we found that phosphorylation of PERK but not of IRE1α stimulates the activation of downstream kinase JNK to directly induce cell death in G1-treated MCF-7 cells. On the other hand, G1-induced p-IRE1α does not activate the XBP1s but rather enhances the activation of CAMKII and is therefore implicated in G1-induced cell death (Figure 7).

To the best of our knowledge, we are the first to reveal a connection where stimulation of GPER by its agonist G1 induced ER Ca^2+^ efflux, leading to ER stress-induced apoptosis in the estrogen receptor-positive MCF-7 cells. Clearly, further studies are needed to demonstrate that this mechanism is also important for other cell lines that show G1-induced cell death and Ca-fluxes. According to Ariazi et al., inhibitors of ER Ca^2+^ channels in the ER membrane, including IP_3_Rs and RyRs, could suppress G1-induced Ca^2+^ efflux in MCF-7 and SKBRr3 cells, respectively [18]. These results are in line with our findings, indicating that the release of Ca^2+^ from the ER via IP_3_Rs and/or RyRs triggers UPR-dependent apoptosis. In this study, we used TG as a positive control, as it is known to block the pumping of cytosolic Ca^2+^ to the ER, leading to ER stress. Similar to G1, TG showed an activation of UPR signaling, and the inhibitor of p-PERK slightly inhibited TG-induced apoptosis. Our previous study of Neuronatin (NNAT), a proteolipid involved in the regulation of Ca^2+^-channels, showed that high expression of NNAT was significantly associated with poor prognosis for overall survival in BC patients [41]. Drugs targeting Ca^2+^ channels/transporters/pumps for cancer treatment have been widely studied in pre-clinical research and even in clinical trials, some compounds of which have shown promising anti-cancer abilities [42]. Therefore, GPER might serve as a potential therapeutic target causing intracellular Ca^2+^ depletion.

Many studies have shown that G1 possesses a strong inhibitory effect on cell growth in GPER-expressing cancer cells [21,22,23,43,44]. Most of these studies report that activation of GPER by its specific agonist G1 inhibits proliferation by inducing cell cycle arrest at G2/M, enhances phosphorylation of histone 3, induces nuclear translocation of ERK1/2 and induces apoptosis by activating caspase 3 and 9. We confirmed that G1 induces cell cycle arrest at G2/M and late apoptosis in MCF-7 cells, which is in line with previous reports. However, we further showed that G1 induces PARP-dependent apoptosis independent of caspase 3 and 9. PARP is one of several cellular substrates of caspases. Cleavage of PARP by caspases is considered to be a hallmark of apoptosis [45]. In addition to caspase 3 and 9, cleavage of PARP could also occur by other caspases like caspase 7 or proteases including cathepsins [46]. Addition of zVAD-fmk blocks PARP processing, but it was reported that zVAD-fmk blocks not only the protease activity of caspases but also that of cathepsins [46]. Further investigations are required to identify the protease responsible for cleavage of PARP in the mechanism of G1-induced non-canonical apoptosis in MCF-7 cells. It should be taken into account that the concentration of G1 used in our and other studies is above the reported IC50 value (3–6 nM). However, G1 showed a specific effect on GPER stimulation indicating in both knockdown of GPER in cancer cell lines which improved cell proliferation and in vivo experiments on tumor xenograft mice models of estrogen receptor (-) BC cells [23], HCT116 CRC [12] and ASG or SNU-216 gastric cancer [44] where tumor size is reduced by G1. Nevertheless, it cannot be completely excluded that other binding proteins contribute to the cell death mechanism at this saturating concentration.

Our results also demonstrated a direct involvement of PERK and IRE1α in G1-induced MCF-7 cell death. GSK2606414 and Kira6, inhibitors of phosphorylation of PERK and IRE1α, respectively, significantly suppressed cell death upon GPER stimulation by G1, indicating ER stress-dependent cell death. However, downstream, activation of CHOP was not involved, but SP600125, a selective inhibitor of JNK-phosphorylation, suppressed G1-induced cell death. It is also known that IRE1α stimulates the activation of the stress kinases JNK and p38 [25]. Under severe ER stress conditions, JNK then eliminates the anti-apoptotic effects of BCL-2 by phosphorylation or leads to the activation of Bax or Bak by phosphorylating Bim, eventually activating cell death via mitochondria [47]. Although we observed an up-regulation of Bax (data not shown), a mitochondrial function is not implicated in G1-induced apoptosis. Notably, in most reports involving Ca^2+^-mediated cell death mechanism, mitochondria play a major role by coupling ER-stress to apoptosis. In general, anti-apoptotic factors like BCL-2 reduce the intracellular ER Ca^2+^ concentration, which, however, is increased by Bax and Bak, eventually leading to the release of cytochrome c to trigger apoptosis [48]. Liu et al. reported that stimulation of GPER by G1 induced apoptosis in CRC is dependent on mitochondria but not on ER stress [12]. Wei et al. also showed that G1 inhibited the growth of estrogen receptor (-) BC and induced mitochondria-related apoptosis [23]. In contrast to these studies, we revealed a different cell death mechanism induced by G1 in MCF-7 cells that is characterized by a new role of PERK in unresolved Ca^2+^ efflux-induced ER stress, where it directly activates the pro-apoptotic JNK protein and independent of mitochondrial function. On the other hand, a recent study by Lee et al. strongly supported our result which showed the dependence of PERK induction in G1-induced cell death in ASG gastic cancer cells and silencing of PERK reduced ER stress signals in the downstream incuding ATF4, CHOP; prevented cell death and increased GPER expression [44]. Moreover, our results might suggest a novel cell death mechanism in eukaryotic cancer cells where JNK is activated directly via p-PERK instead of p-IRE1α. Similar results were observed in a chronic ER stress *Drosophila* model, where PERK/ATF4 activated the JNK pathway through Rac1 and Slpr activation in apoptotic cells, leading to the expression of *Dilp8* and thus to a developmental delay [49]. 

Recent evidence suggests that several signaling pathways are involved in the connection between CAMKII and unresolved ER stress-induced apoptosis. For example, CHOP promotes ER Ca^2+^ release and the subsequent activation of CAMKII downstream the ASK1-JNK signaling cascade, leading to cell death [50]. Otherwise, CAMKII can be the last target in the signaling cascade of CHOP-IRO1α-IP3R1-CAMKII in tunicamycin-induced ER stress [51]. To the best of our knowledge, there is no report on a signaling pathway where CAMKII is directly induced by IRE1α in ER stress-induced cell death. Our results reveal a connection between IRE1α and CAMKII, in which Kira6 inhibits the phosphorylation of CAMKII and G1-induced cell death. It would be interesting to further investigate the role of IRE1α-CAMKII in G1-induced cell death.

Recent research indicates that estrogens and related compounds can also drive cells into apoptosis. HeLa cells undergo apoptosis upon incubation with estrogens E1 and E2 at 10 µM concentration whereas E3, genistein and zearalanone were nearly ineffective [52]. In this case phosphodiesterase 3A was found to be the target molecule and apoptosis was dependent on caspase 9 and the mitochondrial pathway. Interestingly this was mediated by Schlafen 12 which interferes with the protein biosynthesis at the ER. Zearalenone, a *Fusarium*-derived estrogenic mycotoxin causes ER-stress in bovine mammary epithelial cells at 30 µM resulting also in apoptosis via the mitochondrial pathway [53]. For both cases, the proposed apoptotic mechanism is different to the one we propose here for G-1 induced cell death in MCF-7 breast cancer cells. Nevertheless, this clearly shows that the relation between estrogenic compounds and apoptotic cell death needs to be further investigated.

Additional investigations are required to fully address the features and specificities of the stimulation of GPER by its specific agonist G1 in cancer. Our study revealed a connection between ER Ca^2+^ mobilization and the pro-death UPR signaling, triggering MCF-7 cell death when GPER is stimulated by its specific agonist G1. However, our study is limited to one single cell line of estrogen receptor positive MCF-7 BC cells. Further studies using other cell lines would be required to support previous research defining GPER as a target for breast cancer therapy.

## 5. Conclusions

G1-induced Ca^2+^ efflux activates the UPR in MCF-7 cells accompanied by the activation of IRE1α, PERK and ATF6. Although the pro-survival UPR is activated, the pro-death UPR is dominantly stimulated and responsible for G1-induced cell death. Mechanistically, phosphorylation of PERK stimulates the activation of downstream kinase JNK to directly induce apoptosis. On the other hand, G1-induced p-IRE1α does not activate the XBP1s but rather enhances the activation of CAMKII and is therefore also implicated in G1-induced cell death. Our results support previous research promoting GPER as a target for breast cancer therapy.

## Figures and Tables

**Figure 1 biomolecules-09-00503-f001:**
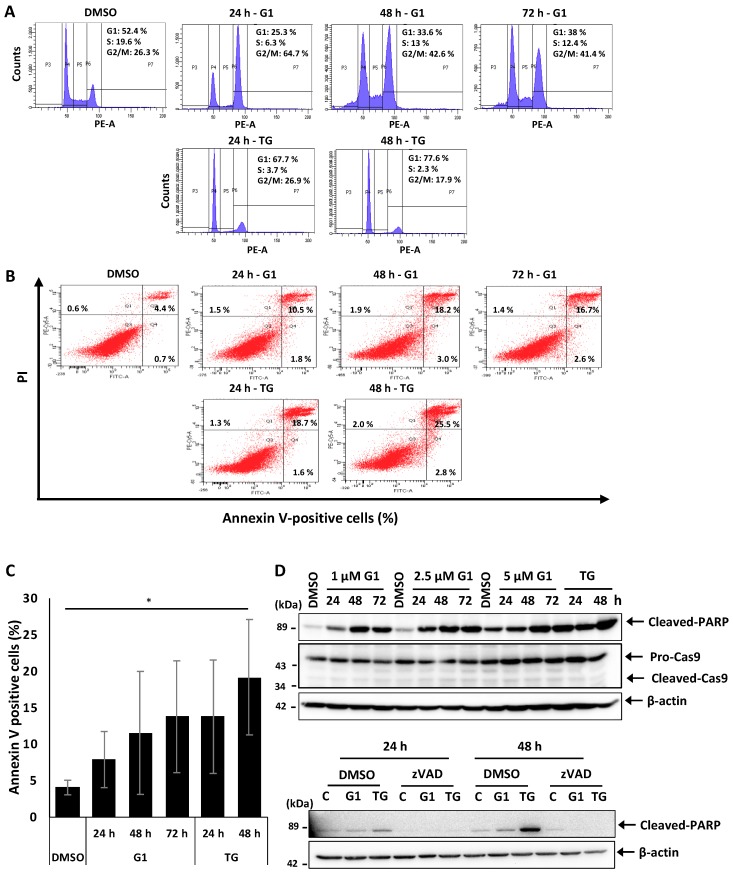
G1 and TG induced cell cycle arrest in G2/M and apoptosis without activating caspase 3 and 9 in MCF-7 cells. MCF-7 cells were treated with 1 μM G1 for 24, 48 and 72 h or with 1 μM TG for 24 and 48 h. (**A**) Cells were stained with propidium iodide (PI), and cell cycle profiles were determined by flow cytometry analysis. The percentage of cells in the different cell cycle phase was calculated. (**B**,**C**) Cells were double stained with PI and Annexin V. Flow cytometry analysis was performed to determine apoptotic and necrotic cells. Representative dot plots and percentage in each quadrant showed cell death undergoing necrosis (Q1), apoptosis (Q4) or late apoptosis (Q2). * *p* < 0.05 when compared with cells treated with vehicle. (**D**) MCF-7 cells were treated with 1, 2.5 and 5 μM G1 for 24, 48 and 72 h, or with 1 μM TG for 24 and 48 h, or pretreated with 20 μM pan caspases inhibitor zVAD-fmk for 1 h and then treated with 1 μM G1 or TG for 24 and 48 h. Total protein lysates were subjected to Western blotting using appropriate antibodies as indicated.

**Figure 2 biomolecules-09-00503-f002:**
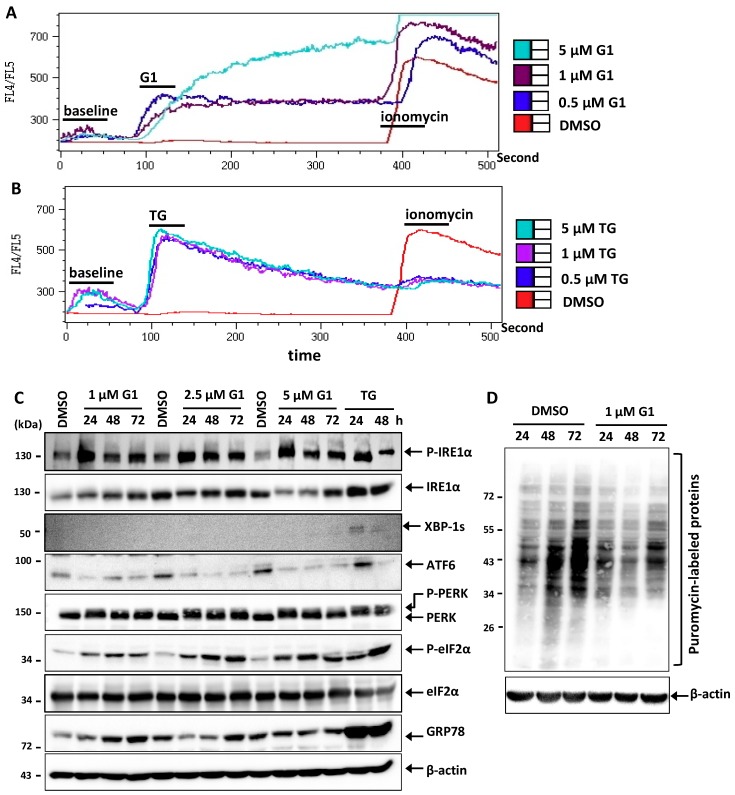
G1 and TG induced Ca^2+^ efflux and activated the UPR in MCF-7 cells. (**A**,**B**) Intracellular Ca^2+^ concentration was measured by flow cytometry in Indo-1-stained MCF-7 cells after stimulation with different concentrations of G1 or TG. To demonstrate successful loading with the dye, we induced maximal Ca^2+^ release by adding calcium ionophore ionomycin (10 mg/mL) at the end of a measurement. (**C**) MCF-7 cells were treated with 1, 2.5 and 5 μM G1 for 24, 48 and 72 h, or with 1 μM TG for 24 and 48 h. (**D**) MCF-7 cells were treated with 1 μM G1 for 24, 48 and 72 h. A total of 15 min before the end of a treatment time, 10 μg/mL puromycin was added. Total protein lysates were subjected to Western blotting using appropriate antibodies as indicated.

**Figure 3 biomolecules-09-00503-f003:**
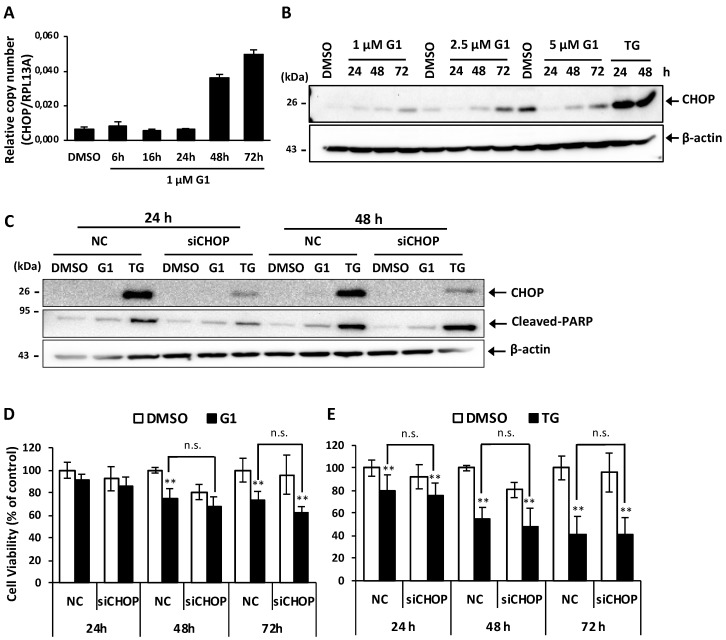
Up-regulation of CHOP expression was not involved in G1-induced cell death in MCF-7 cells. MCF-7 cells were treated with 1, 2.5 and 5 μM G1 at different time points. Total mRNA (**A**) and protein lysates (**B**) were isolated, and the levels of the expression of CHOP-mRNA were measured by real-time PCR and normalized to RPL13A and Western blotting. MCF-7 cells were transfected with siRNA-CHOP or negative control (NC) for 24 h, and then the cells were challenged with 1 μM G1 or 1 μM TG for 24, 48 and 72 h. (**C**) Total protein lysates were subjected to Western blotting using appropriate antibodies. (**D**,**E**) Cell viability was measured by the LDH activity assay. ** *p* < 0.01 when compared with cells treated with vehicle.

**Figure 4 biomolecules-09-00503-f004:**
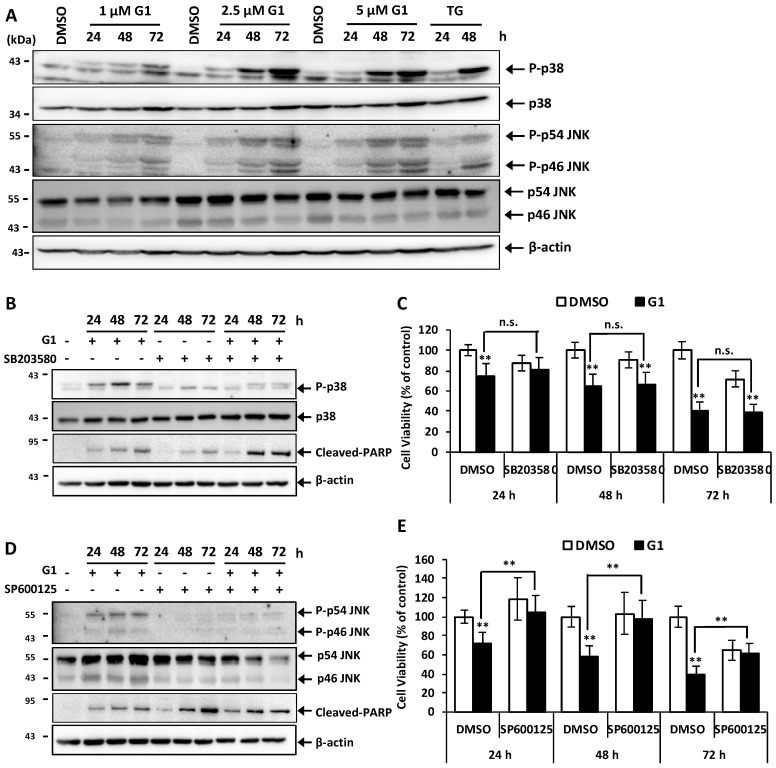
G1 activated p38 and JNK and a selective inhibitor of phosphorylation of JNK inhibited G1-induced cell death in MCF-7 cells. (**A**) MCF-7 cells were treated with 1, 2.5 and 5 μM G1 for 24, 48 and 72 h, or with 1 μM TG for 24 and 48 h. Total protein lysates were subjected to Western blotting using appropriate antibodies as indicated. Cells were pretreated with 1 μM SB203580, a selective inhibitor of p38 MAP kinase or 10 μM SP600125, a selective inhibitor of JNK kinase and then exposed to 1 μM G1 for 24, 48 and 72 h. (**B**,**D**) Total protein lysates were subjected to Western blotting using appropriate antibodies as indicated. (**C**,**E**) Cell viability was measured by the LDH activity assay. ** *p* < 0.01 when compared with cells treated with vehicle or among G1-treated groups. + is indicated for the presence of a treatment meanwhile – is indicated for the absence of a treatment.

**Figure 5 biomolecules-09-00503-f005:**
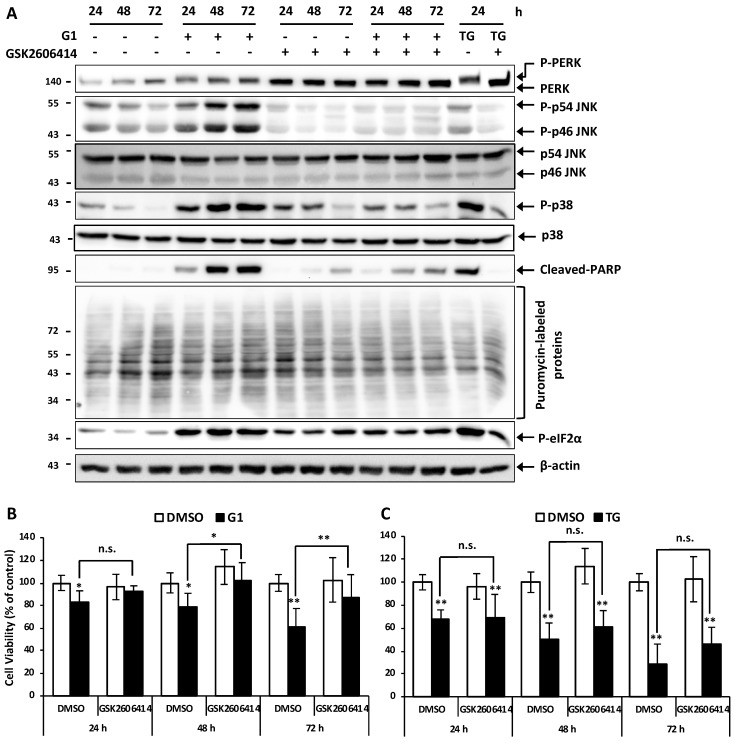
Inhibition of PERK inhibited the activation of JNK and prevented G1-induced cell death in MCF-7 cells. MCF-7 cells were pretreated with 5 μM GSK2606414, an inhibitor of PERK kinase for 1 h and then exposed to 1 μM G1 or TG for 24, 48 and 72 h. (**A**) Total protein lysates were subjected to Western blotting using appropriate antibodies as indicated. (**B**,**C**) Cell viability was measured by the resazurin assay. * *p* < 0.05, and ** *p* < 0.01 when compared with cells treated with vehicle or among G1-treated groups. + is indicated for the presence of a treatment meanwhile – is indicated for the absence of a treatment.

**Figure 6 biomolecules-09-00503-f006:**
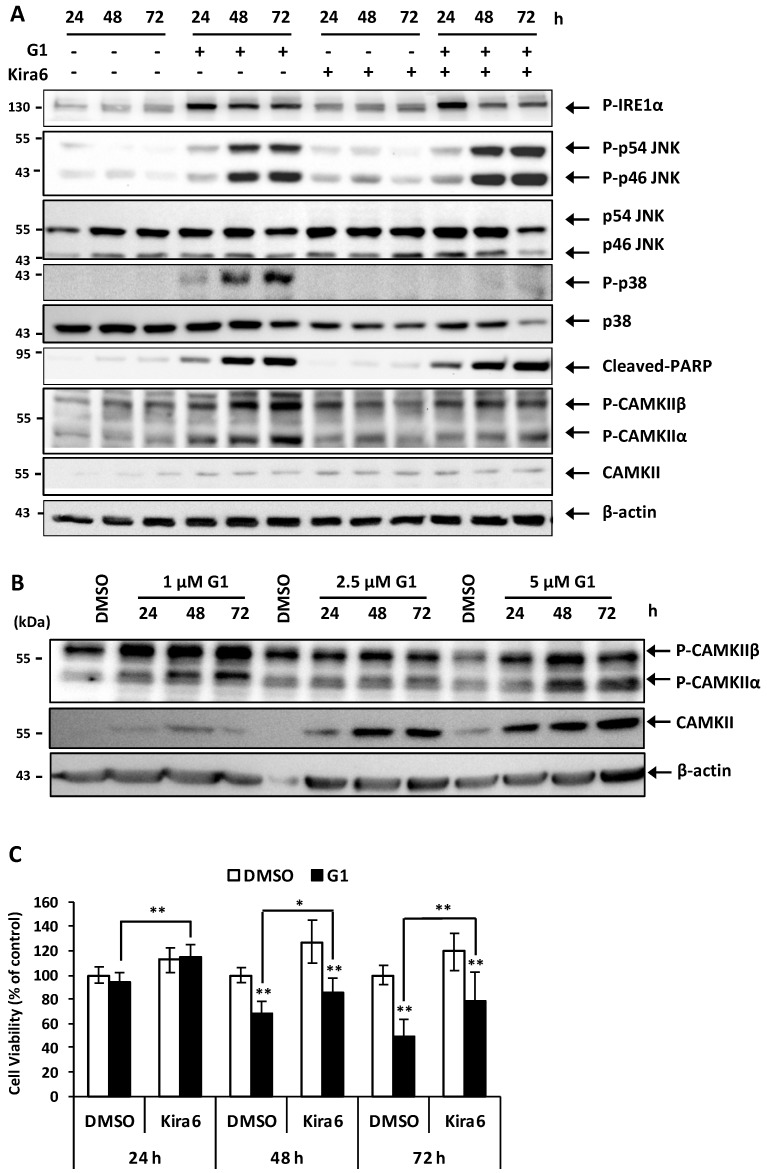
Inhibition of IRE1α inhibited the phosphorylation of CAMKII and prevented G1-induced cell death in MCF-7 cells. MCF-7 cells were pretreated with 0.05 μM Kira6, an inhibitor of IRE1α for 1 h and then exposed to 1 μM G1 for 24, 48 and 72 h. (**A**)Total protein lysates were subjected to Western blotting using appropriate antibodies as indicated. (**C**) Cell viability was measured by the resazurin assay. (**B**) MCF-7 cells were treated with 1, 2.5 and 5 μM G1 for 24, 48 and 72 h. Total protein lysates were subjected to Western blotting using appropriate antibodies as indicated. * *p* < 0.05, and ** *p* < 0.01 when compared with cells treated with vehicle or among G1-treated groups. + is indicated for the presence of a treatment meanwhile – is indicated for the absence of a treatment.

**Figure 7 biomolecules-09-00503-f007:**
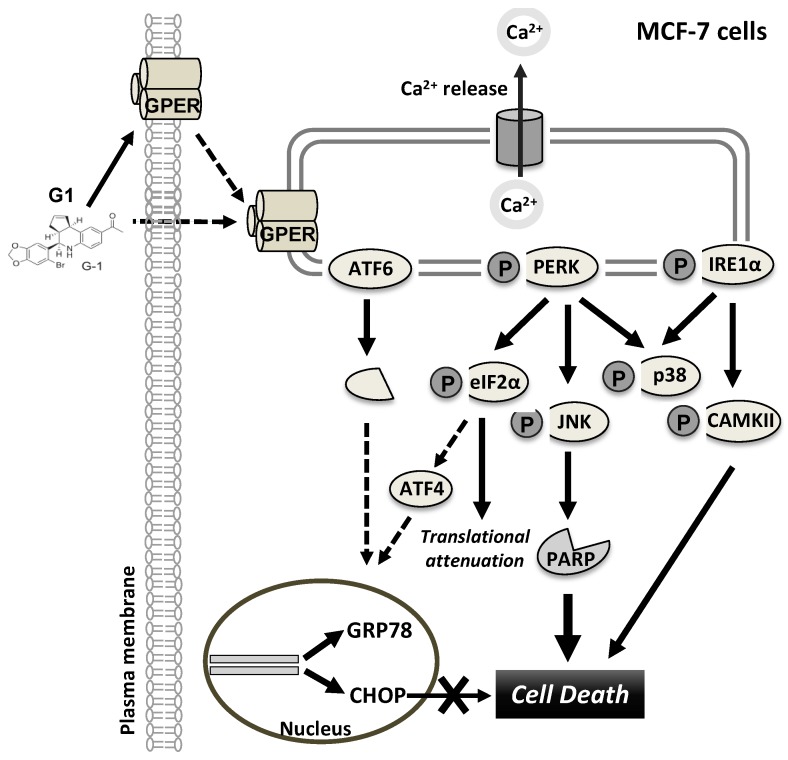
Model of the proposed mechanism of G1-induced cell death. G1 leads to a decrease of ER Ca^2+^ stores in MCF-7 cells, and this event activates the UPR in MCF-7 cells, including the activation of IRE1α, PERK and ATF6. The pro-survival UPR is activated exhibiting upregulation of GRP78 to assist the proper folding capacity of the ER and eIF2α-phosphorylation to reduce translation of nascent proteins. However, pro-death UPR signaling plays a major role in G1-induced ER stress, which, in turn, regulates cell death. CHOP is upregulated but not involved in the G1-induced cell death machinery. In contrast, phosphorylation of both PERK and IRE1α is implicated in G1-induced cell death. Phosphorylation of PERK stimulates the activation of downstream kinase JNK to directly induce apoptosis, while phosphorylation of IRE1α induces phosphorylation of CAMKIII.

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
