# Peer review of "G-Protein-Coupled Estrogen Receptor (GPER)-Specific Agonist G1 Induces ER Stress Leading to Cell Death in MCF-7 Cells"

_biomolecules, 2019, doi:10.3390/biom9090503_

Round 1
Reviewer 1 Report
The authors studied mechanisms in cell death induced by activation of G-protein coupled estrogen receptor. Using MCF7, a ER+ breast cancer cell line, the authors demonstrated how specific activation of GPER by G1 increases intracellular calcium concentration via release of endoplasmic reticulum calcium. They provided evidence showing three signaling molecules (IRE1a, PERK, ATF6) in pro-death UPR signaling induced by ER stress. It is interesting that authors discovered two mechanisms that mediate in G1-induced cell death in ER+ breast cancer cells: a new role of CAMKII and PARP-dependent apoptosis independent of caspase-3 and 9. Overall, it's a good manuscript for Biomolecules journal.
Minor:
(1) Line 283: space after p<0.05
(2) Line 349: Figure 2A, 2B legend, no description of ionomycin, although ionomycin is a common calcium ionophore.
Author Response
The authors studied mechanisms in cell death induced by activation of G-protein coupled estrogen receptor. Using MCF7, an ER+ breast cancer cell line, the authors demonstrated how specific activation of GPER by G1 increases intracellular calcium concentration via release of endoplasmic reticulum calcium. They provided evidence showing three signaling molecules (IRE1a, PERK, ATF6) in pro-death UPR signaling induced by ER stress. It is interesting that authors discovered two mechanisms that mediate in G1-induced cell death in ER+ breast cancer cells: a new role of CAMKII and PARP-dependent apoptosis independent of caspase-3 and 9. Overall, it's a good manuscript for Biomolecules journal.
Minor:
(1) Line 283: space after p<0.05
(2) Line 349: Figure 2A, 2B legend, no description of ionomycin, although ionomycin is a common calcium ionophore.
We thank the reviewer for these encouraging remarks and have revised the text according to the minor points.
Reviewer 2 Report
In this manuscript by Vo et al., GPER agonist G1-mediated cell death was investigated. The authors have found that G1 induces ER stress and thus death of MCF-7 cells. The experimental design and results of this manuscript are well organized. However, there are a few things the authors need to address.
The authors have used only one cell line (MCF-7). The authors should consider other ER positive breast cancer cell lines, such as T47D, to support their hypothesis. The authors found that the amount of late apoptotic cells were continuously increased at 48 and 72 h. However, FACS data is conflicted with their explanation in figure 1B (72 h-G1). The authors suggest that G1 induced apoptosis is independent with Caspase-3 and -9 activation. Also, treatment of zVAD-fmk was reduced PAPR cleavage without recovery of cell viability. For these reasons, cleaved-PARP appears to be inappropriate as a cell death marker in this study. What about consider another type of cell death or other factors such as AIF. (doi: 10.1074/jbc.M109.044206) In this study, p-PERK was examined using PERK antibody. The authors need to use p-PERK antibody to determine the level of PERK phosphorylation. In Fig. 2C, TG treatment promotes phosphorylation of IRE1a. However, TG does not affect IRE1a phosphorylation in Fig. 6a.Author Response
In this manuscript by Vo et al., GPER agonist G1-mediated cell death was investigated. The authors have found that G1 induces ER stress and thus death of MCF-7 cells. The experimental design and results of this manuscript are well organized. However, there are a few things the authors need to address.
The authors have used only one cell line (MCF-7). The authors should consider other ER positive breast cancer cell lines, such as T47D, to support their hypothesis. The authors found that the amount of late apoptotic cells were continuously increased at 48 and 72 h. However, FACS data is conflicted with their explanation in figure 1B (72 h-G1). The authors suggest that G1 induced apoptosis is independent with Caspase-3 and -9 activation. Also, treatment of zVAD-fmk was reduced PAPR cleavage without recovery of cell viability. For these reasons, cleaved-PARP appears to be inappropriate as a cell death marker in this study. What about consider another type of cell death or other factors such as AIF. (doi: 10.1074/jbc.M109.044206) In this study, p-PERK was examined using PERK antibody. The authors need to use p-PERK antibody to determine the level of PERK phosphorylation. In Fig. 2C, TG treatment promotes phosphorylation of IRE1a. However, TG does not affect IRE1a phosphorylation in Fig. 6a.
The authors have used only one cell line (MCF-7). The authors should consider other ER positive breast cancer cell lines, such as T47D, to support their hypothesis.
It is indeed important to show whether a mechanisms has a more general importance by investigating several independent cell lines. It will, however, not be too surprising to find that individual cell lines might respond to a single stimulus by using diverse signaling and cell death pathways. That G1-induced cell death (apoptosis) indeed occurs in several GPER-expressing breast cancer (ER-positive and also ER-negative) cell lines has been shown and published by our collaboration partners from the clinic of gynecology and other groups before (I.e. EC50 for cell death in MCF-7 was 1.3 µM, SK-BR3: 2.8 µM, MDA-MB 231: 3.2µM and MDA-MB-468: 0.4 µM). We have also data showing that Ca-fluxes are elicited by G1 in SKBR3 and MDA-MB 231.
Here however, we focused on the MCF-7 cell line, which is somehow special as these cells do not express caspase 3 and this feature could define a specific cell death mechanism. Other cell lines, for instance SkBr3 (ER-negative, HER2 upregulated) or MDA-MD-231 (triple-negative) showed cleavage of caspase 3 and upregulation of p21 by cross-talk of GPR30/EGFR and p53 was involved in G-1-induced ER− breast cancer cell growth arrest [1]. In the case of the colorectal cancer cell line, HCt116 cell, caspase 3 was also cleaved and ROS/ERK1/2 signals were involved in suppression effects of G-1 on CRC cell growth [4].
The authors found that the amount of late apoptotic cells were continuously increased at 48 and 72 h. However, FACS data is conflicted with their explanation in figure 1B (72 h-G1).
It is indeed true that the flow cytometry did not really show a constant increase in late apoptotic cells until 72 h. However, we observed an increased number of detached cells and cellular debris with increasing time, which we believe can be attributed to ongoing apoptosis. These cells have been lost during the sample preparation for flow cytometry and the flow cytometry showed a more or less constant steady state rate of apoptosis.
Therefore, we stated that late apoptotic cells increased with time. To make this clear, we have introduced changes in the manuscript clarifying this point.
The authors suggest that G1 induced apoptosis is independent with Caspase-3 and -9 activation. Also, treatment of zVAD-fmk was reduced PAPR cleavage without recovery of cell viability. For these reasons, cleaved-PARP appears to be inappropriate as a cell death marker in this study. What about consider another type of cell death or other factors such as AIF. (doi: 10.1074/jbc.M109.044206)
We thank the reviewer for this interesting hint. Indeed, we did not investigate thoroughly whether PARP cleavage (or AIF translocation) would be necessary for cell death. Although the caspase inhibitor zVAD-fmk prevented PARP cleavage, we observed severe cell death induced by zVAD-fmk alone, which we attributed to the intracellular release of toxic fluoroacetate (Ref 36). So these experiments were not suitable to draw conclusions on the cleavage of PARP and cell death. Nevertheless, in the other inhibitor experiments the degree of PARP cleavage was consistent with the effects on cell death. Because caspase (3/7) activity or caspase 9 activation by cleavage could not be observed upon G-1 treatment, we did not investigate this further. Nevertheless, we agree with the reviewer that G1 induced apoptosis is independent of caspase-3 and -9 and therefore other proteases such as cathepsins are supposed to cleave PARP. Cathepsins can also be inhibited by zVAD-fmk. We also mentioned above that G1 could induce more than one cell death mechanisms, independently via PERK/JNK phosphorylation and IRE1a/CAMKII phosphorylation. We further showed that G1 induced Bax expression but was mitochondria-independent.
In this study, p-PERK was examined using PERK antibody. The authors need to use p-PERK antibody to determine the level of PERK phosphorylation.
One of us (DKV) has previously published a study showing that the band shift technique is reliable (Urano et al. Cell Death Discovery (2019) 5:113 https://doi.org/10.1038/s41420-019-0192-4) and several other studies have applied the same technique (Su Q, et al. J Biol Chem. 2008 Jan 4;283(1):469-75 and Koumenis C, et al. Mol Cell Biol. 2002 Nov;22(21):7405-16.)
So we could show a band shift of PERK in G1- or TG-treated cells and at the same time no band shift was observed in control group. As this band shift was clearly visible in our experiments, we decided that using a p-PERK antibody would not be necessary. Additionally, as the affinity of this polyclonal antibody for the phosphorylated and non-phosphorylated form of this protein is presumably identical, the technique allows a direct comparison of the band intensities.
In Fig. 2C, TG treatment promotes phosphorylation of IRE1a. However, TG does not affect IRE1a phosphorylation in Fig. 6a.
We thank the referee for this remark. Indeed, the TG effect was not really evident in the particular Western blot experiment that we used for this figure. We originally intended to show the whole data set of one single representative experiment in this figure and did not really care for the problem with the TG. As the TG treatment does not affect our interpretation in clarifying the involvement of IRE1a in G1-induced cell death, we decided to remove this data of TG treatment from Fig. 6a. Nevertheless, TG caused phosphorylation of IRE1a in the majority of our experiments.
Reviewer 3 Report
This manuscript addresses the molecular mechanism of GPER-induced death due to ER-stress. This paper should be rejected in its current form due to serious flaws in experimental design.
Major concerns:
The authors use a single dose for the synthetic GPER agonist, G1 (1 uM) that exceeds the reported Ki for its receptor (3-6 nM) by 250-500-fold. Critical controls for agonist specificity. Related steroid hormones of similar structure with similar Kd's (ie 17b-estradiol) as well as negative controls that do not demonstrate receptor binding or signaling (cortisol, aldosterone, etc) tested at various doses between the Kd and the dose selected by the author (500-fold above reported Ki). Critical controls for receptor specificity. Pharmacological inhibitors AND genetic knockdown of the receptor (iRNA or CRISPR-Cas9). All of the data are established in a SINGLE cell line.In addition, the authors have made blanket statements regarding the state of GPER in cancer biology (that it functions as a tumor suppressor) without discussing an equal number of papers that show a direct association between GPER and cancer progression. This biases the reader further towards the significance of the authors suggested molecular mechanism of drug-induced death. If in fact, G1 induces death by ER-stress, it is much more significant to show this in the context of a concentration-response curve.
Author Response
This manuscript addresses the molecular mechanism of GPER-induced death due to ER-stress. This paper should be rejected in its current form due to serious flaws in experimental design.
Major concerns:
The authors use a single dose for the synthetic GPER agonist, G1 (1 uM) that exceeds the reported Ki for its receptor (3-6 nM) by 250-500-fold. Critical controls for agonist specificity. Related steroid hormones of similar structure with similar Kd's (ie 17b-estradiol) as well as negative controls that do not demonstrate receptor binding or signaling (cortisol, aldosterone, etc) tested at various doses between the Kd and the dose selected by the author (500-fold above reported Ki). Critical controls for receptor specificity. Pharmacological inhibitors AND genetic knockdown of the receptor (siRNA or CRISPR-Cas9). All of the data are established in a SINGLE cell line.
In addition, the authors have made blanket statements regarding the state of GPER in cancer biology (that it functions as a tumor suppressor) without discussing an equal number of papers that show a direct association between GPER and cancer progression. This biases the reader further towards the significance of the authors suggested molecular mechanism of drug-induced death. If in fact, G1 induces death by ER-stress, it is much more significant to show this in the context of a concentration-response curve.
We explained above (comments to the editor) that 1 µM G1 is a concentration that has been applied in many published studies on G1 induced cell death before and also tried to explain why we decided to choose this high concentration in our experiments. However, we agree that this dosage is certainly close to, if not above saturation. Nevertheless other authors could show by siRNA experiments or by using the specific antagonist G15 that G1-induced cell death indeed depends on GPER. This does not mean that low concentrations of G1 can have opposing effects such as stimulating proliferation. Also, Scaling et al (Ref 39) clearly stated that “… in vitro studies have demonstrated that the G protein-coupled estrogen receptor /…/ can modulate proliferation in breast cancer cells both positively and negatively depending on cellular context”.
In the Bologa et al publication where G-1 has been introduced (ref 17) G-1 induced Ca-fluxes were already detectable at concentrations of 1 nM and saturation was achieved at concentrations of about 0.5 µM. These experiments were done with COS7 cells, artificially expressing GPER-GFP fusions. In our experiments, we needed much higher concentrations to obtain Ca-fluxes. We think the differences in the dose response curve for calcium fluxes between our study and this publication might reside in the expression level of GPER or the sensitivity of our detection method. However, Bologa et al also described an effect of G1 on migration of MCF-7 and SKBR3. These effects became already detectable at about 10-11 M and increased until the effect became saturated at 10-6 M, which is the concentration we used in this study.
In order to achieve a more balanced view, we have now added a statement about the high G-1 concentrations needed to reduce viability/proliferation of cell culture models to the discussion.
We also like to thank the referee for the critic on our statement on GPER as a potential “tumor suppressor”. We agree that this remark was much too harsh in the context of the existing data and we consequently removed this statement from the abstract and the manuscript. We agree that the role of GPER in cancer biology is still not completely understood and we hope that we were able to make this now more clear in the manuscript. GPER stimulation by estrogen (or tamoxifen) indeed results in proliferation, this is well known and we mention this in the introduction as well as in the discussion. I.e., we have explained on page 2, line 58 that “high GPER expression also correlated with the development of distant metastasis in BC [13]. In case of BC anti-estrogen therapy, high GPER levels were associated with a shorter disease-free survival under tamoxifen compared to those receiving aromatase inhibitors [14]. These conflicting observations are in line with the idea that non-specific GPER agonists, such as estrogen and tamoxifen, induce cell proliferation by GPER stimulation, thereby replacing the tamoxifen-blocked genomic estrogen signaling and activating a cross talk of GPER with growth factor signaling [15,16].”
We also agree that experiments with a single cell line cannot prove the general importance of a certain mechanism. Indeed, we believe that MCF-7 might be special, as an important caspase (cas3) is not expressed in these cells. So we have added this to the discussion, now clearly stating that further experiments with other cell lines are needed to claim that the described mechanism has a more general meaning.
Round 2
Reviewer 2 Report
The authors have addressed suggested concerns.
Author Response
Thank you very much for your positive reply.
Reviewer 3 Report
The authors did not address the concerns that were levied against this paper by performing additional experiments. Instead, they simply pointed out examples in the literature where others have also used exceedingly high doses of the syntheticagonist G1. Critical to answer is the question as to “why is this agonist toxic? And are similar ERAD-associated death signals observed with high doses of other estrogen-related compounds? Again, the authors use a SINGLE cell line and it is insufficient to simply state in the Discussion that the primary message of the paper should be taken into context based on this limitation. Furthermore, the importance of addressing multiple cell lines in this paper is critical to evaluate the molecular mechanism of cell death by G1 reported here. It is also concerning that the authors propose that G1 may be used as a therapeutic while there is no known reported information regarding the concentrations of G1 that can be achieved in vivo, and conservatively it is likely that the high dose that is used (1 uM) may not be achieved. In addition, no evidence that the effects that are measured are not “off-target” effects. RNAi or gene knockout experiments are critical to conduct to demonstrate receptor specificity. It is insufficient to reference work by others as a reason for failing to include these experiments here.
